# Helicobacter Species and Hepato-Biliary Tract Malignancies: A Systematic Review and Meta-Analysis

**DOI:** 10.3390/cancers15030595

**Published:** 2023-01-18

**Authors:** Beatriz Gros, Alberto Gómez Pérez, María Pleguezuelo, Francisco Javier Serrano Ruiz, Manuel de la Mata, Manuel Rodríguez-Perálvarez

**Affiliations:** 1Department of Gastroenterology and Hepatology, Hospital Universitario Reina Sofía, 14004 Córdoba, Spain; 2Maimonides Institute of Biomedical Research (IMIBIC), 14004 Córdoba, Spain; 3Centro de Investigación Biomédica en Red de Enfermedades Hepáticas y Digestivas (CIBERehd), 28029 Madrid, Spain

**Keywords:** *Helicobacter*, cholangiocarcinoma, cancer, neoplasms

## Abstract

**Simple Summary:**

Chronic infections are major drivers of cancer. *Helicobacter species* is one of the most established pro-oncogenic pathogens, but its relationship with hepatobiliary tract malignancies is controversial. We performed a systematic review and meta-analysis of 26 observational studies, including 1203 patients with hepatobiliary tract malignancies and 2880 controls. *Helicobacter species* chronic infection demonstrated by direct microbiological methods tripled the risk of hepatobiliary tract malignancies, and this effect was consistent for different types of specimens tested, including bile, gastric tissue, hepatic/biliary tissue, and serum. However, available studies were heterogeneous, and the overall quality of the evidence was low, translating into a grade of uncertainty. Prospective studies, randomized, if possible, are required to delineate interventions at a public healthcare scale, particularly in geographic areas with increased incidence of hepatobiliary tumors.

**Abstract:**

*Helicobacter species* may cause chronic inflammation of the biliary tract, but its relationship with cancer is controversial. We performed a systematic review and meta-analysis to evaluate the association between *Helicobacter species* and hepatobiliary tract malignancies. Twenty-six studies (4083 patients) were included in qualitative synthesis, and 18 studies (*n* = 1895 qualified for meta-analysis. All studies were at high-intermediate risk of bias. Most studies combined several direct microbiological methods, mostly PCR (23 studies), culture (8 studies), and/or CLOtest (5 studies). Different specimens alone or in combination were investigated, most frequently bile (16 studies), serum (7 studies), liver/biliary tissue (8 studies), and gastric tissue (3 studies). Patients with *Helicobacter species* infection had an increased risk of hepatobiliary tract malignancies (OR = 3.61 [95% CI 2.18–6.00]; *p* < 0.0001), with high heterogeneity in the analysis (I^2^ = 61%; *p* = 0.0003). This effect was consistent when *Helicobacter* was assessed in bile (OR = 3.57 [95% CI 1.73–7.39]; *p* = 0.0006), gastric tissue (OR = 42.63 [95% CI 5.25–346.24]; *p* = 0.0004), liver/biliary tissue (OR = 4.92 [95% CI 1.90–12.76]; *p* = 0.001) and serum (OR = 1.38 [95% CI 1.00–1.90]; *p* = 0.05). Heterogeneity was reduced in these sub-analyses (I^2^ = 0–27%; *p* = ns), except for liver/biliary tissue (I^2^ = 57%; *p* = 0.02). In conclusion, based on low-certainty data, *Helicobacter species* chronic infection is associated with a tripled risk of hepatobiliary tract malignancy. Prospective studies are required to delineate public health interventions.

## 1. Introduction

Several chronic infections are well-known drivers of cancer in humans [1]. *Helicobacter pylori (HP)*, *Opisthorchis viverrini*, *Clonorchis sinensis*, and hepatitis B and C viruses, among other pathogens, have been classified as group 1 carcinogens [2]. In 2018, 2.2 million cancers were attributed to chronic infections worldwide, among which HP was paramount, being responsible for 810.000 newly diagnosed malignancies (mainly non-cardia gastric adenocarcinoma) [3].

Primary liver cancer, which includes mainly hepatocellular carcinoma (HCC) (75–85%) and intrahepatic cholangiocarcinoma (ICC) (10–15%), ranks the sixth malignancy worldwide in terms of prevalence. Indeed, primary liver tumors account for 4.7% of all incident cancers, being the third cause of cancer-related death [4]. Pancreatic cancer and gallbladder cancer represent 2.6% and 0.9% of new cancer diagnoses, respectively, being both characterized by their lethality [5].

There is controversy regarding the epidemiological association between HP (and other *Helicobacter* species infection) and hepatobiliary tract malignancies [6]. The available evidence relies mostly on case-control studies from Asia, where the highest incidence of these malignancies has been reported [7,8]. The interpretation of observational studies is often challenging due to the high prevalence of other confounding pro-oncogenic infections, such as liver fluke and viral hepatitis [9]. Included cohorts are usually small, and healthy controls are lacking for obvious ethical reasons [10]. 

The unequivocal assessment of the attributable pro-oncogenic risk of *Helicobacter* spp. is paramount to delineating public health policies. These data would allow us to identify groups of risk for potential screening programs and to implement therapeutic interventions at a population level. The present systematic review and meta-analysis aim to summarize the available evidence regarding *Helicobacter* spp. and hepatobiliary tract malignancies with a focus on possible prevention strategies.

## 2. Materials and Methods

### 2.1. Search Strategy

We searched MEDLINE, Cochrane Controlled Trial Register (CENTRAL), EMBASE, and Science Citation Index databases, without language restrictions, from inception to May 2022. In addition, the recent literature reviews on the topic, if available, were hand-searched for additional relevant references. We identified studies by using different combinations of the following keywords or equivalent free-text terms: [“helicobacter” OR “helicobacter mustelae” OR “helicobacter hepaticus” OR “helicobacter felis” OR “helicobacter heilmannii” OR “Helicobacter species”] AND [“biliary tract neoplasms” OR “malignancy” OR “cholangiocarcinoma” OR “tumor” OR “cancer”]. 

### 2.2. Selection Criteria

The Preferred Reporting Items for Systematic reviews and Meta-Analyses (PRISMA) statement was used to illustrate the flowchart of included and excluded studies [11]. Randomized clinical trials, cohort studies, or case-control studies published in full length comparing the incidence or prevalence of biliary tract malignancies according to the presence of Helicobacter species assessed by direct microbiological methods were included. Exclusion criteria were as follows: studies performed in children or adolescents (<18 years old), uncontrolled studies (mainly case series), editorials or the literature reviews, and studies performed in animal models and in vitro experiments. Studies were also excluded if the approval from an ethics research committee was not disclosed or if the provided data were insufficient to weigh the risk of cancer attributable to Helicobacter species after contact the corresponding author. Two review authors (BG and AGP) independently screened the retrieved references by examining the title, abstract and full texts whenever necessary. Any discrepancy was resolved initially by discussion. Persistent disagreements were resolved by a third review author (MR-P). If there was any doubt as to whether two studies shared the same participants, completely or partially (by identifying common authors, institutions, and overlapping recruiting periods), we contacted the study authors to clarify whether the study report was duplicated. In the absence of a satisfying response, only the study with the larger sample size was considered.

### 2.3. Data Extraction

Two review authors (BG and AGP) performed the data extraction independently, using a predefined chart specifically designed for this study, with disagreements solved by a third author (MR-P). Data extracted from each study included first author, country and year of publication, sample size, sample collection, method of *Helicobacter* detection, diagnosis of cases and controls, matching strategy, patient characteristics, type of specimen, organisms identified, number of positive cases for *Helicobacter species* in each group, and adjusted odds ratios if provided. The corresponding author of each study was contacted in case of relevant missing information. If available, we obtained separate data for participants with different types of biliary malignancies.

### 2.4. Statistical Analysis and Risk of Bias Assessment

Studies providing sufficient data were included for quantitative synthesis (meta-analysis). We calculated the odds ratio with a 95% credible interval (or Bayesian confidence interval) to estimate the probability of hepatobiliary tract malignancies according to the presence or absence of Helicobacter infection. Since we expected significant clinical heterogeneity due to different methodologies for helicobacter assessment and different types of malignancies, random effects models were implemented. I^2^ values were used as indicators of the degree of intertrial heterogeneity. The possibility of publication bias was explored with funnel plots. The quality of the evidence was assessed using the checklist provided by the Clarity group at McMaster University, which evaluates the risk of bias in case-control studies by assessing five study domains as follows: quality of assessment of exposure, confidence that cases had developed the outcome of interest but not controls, appropriate cases selection, appropriate control selection and appropriate case-control matching according to important prognostic variables or statistical adjustment. Each of these domains is classified into four categories: low risk of bias, probably low risk of bias, probably high risk of bias, or definitely high risk of bias [12]. Analyses were performed using RevMan version 5.1 (Nordic Cochrane Centre, Copenhagen, Denmark). This systematic review was registered in PROSPERO (registration number 42022348410).

## 3. Results

### 3.1. Description of the Literature Search Strategy

A total of 712 references were obtained from the literature search. The PRISMA flowchart depicting the study selection is shown in Figure 1. After removing duplicates (*n* = 66), 645 records were screened. Six hundred and seventeen references were excluded because they referred to animal models (*n* = 31), were the literature reviews, editorials, or case records (*n* = 60), or were unrelated to the topic of interest (*n* = 526). Two studies, including an identical number of patients from the same institution and with a widely overlapping enrolment period, were combined to avoid data duplication. Finally, twenty-six articles were included for qualitative synthesis.

### 3.2. Search Results and Study Characteristics

The main characteristics of the included studies are summarised in Table 1. All studies had a retrospective case-control design and were published between 1995 and 2022. In all, 4083 participants were included, comprising 1203 patients with hepatobiliary tract malignancies, 532 patients with benign hepatobiliary abnormalities, 633 patients with hepatobiliary abnormalities (without differentiating between benign and malignant conditions in three studies), and 1715 healthy controls. Sample size ranged from *n* = 7 to *n* = 551. The types of cancers were cholangiocarcinoma (*n* = 438; 36.4%), hepatocellular carcinoma (*n* = 231; 19.2%), gallbladder cancer (*n* = 111, 9.2%), pancreatic cancer (*n* = 96, 8%), ampulla of Vater cancer (*n* = 45, 3.7%) colangio-hepatocellular carcinoma (*n* = 2, 0.2%), cystoadenocarcinoma (*n* = 1, 0.1%), other bile tract malignancies (*n* = 279, 23.2%). Eight studies reported “bile tract cancer” comprising gallbladder and bile duct cancer without providing individual data for each tumor location. Participants acting as controls underwent imaging techniques to rule out hepatobiliary malignancies, and they had most frequently other non-malignant biliary diseases such as O. Viverrini negative controls (*n* = 382, 22.3%), benign biliary diseases (*n* = 204, 11.9%), gallstones or bile duct stones (*n* = 194, 11.3%), chronic cholecystitis (*n* = 90, 5.3%) or other less prevalent biliary diseases such as Caroli disease, primary sclerosing cholangitis, gallbladder polyps, and acute pancreatitis, among others. Of note, 638 (37.2%) patients in the control group were matched controls of large cohort studies. There was a predominance of studies from Asia (Japan [*n* = 6], Thailand [*n* = 4], Japan and Thailand [*n* = 1], Korea [*n* = 2], Taiwan [*n* = 1], Pakistan [*n* = 1], India [*n* = 1], Iran [*n* = 1]), followed by Europe (Germany [*n* = 2], Yugoslavia [*n* = 1], Finland [*n* = 1], France [*n* = 1], Sweden [*n* = 1]), South America (Mexico [*n* = 2]), and North America (Canada [*n* = 1]). The age and gender of the study participants were not disaggregated for cases and controls in 14 and 11 studies, respectively. For the remaining references, the pooled average age was 56.01 years in cases and 54.47 years in controls, the prevalence of women being 51.4% among cases and 47.9% among controls.

### 3.3. Helicobacter spp. Isolation

As for the microbiological detection methods, the majority of the studies used a single methodology (*n* = 15). *Helicobacter* spp. chronic infection was confirmed by means of PCR (*n* = 23), culture (n = 8), CLOtest (*n* = 5), histological and/or immunohistochemical (*n* = 3), ELISA (*n* = 4), Western Blot (*n* = 1) and multiplex assay (*n* = 1). In studies using PCR, the most frequently primer genes were 16rRNA 17/20 (85%) (either to *Helicobacter genus* or specific *Helicobacter specie*), urease A 8/20 (40%), cytotoxin-associated gene A (CagA) 7/20 (35%), Vacuolating cytotoxin A (VacA) 6/20 (30%) and glmM gene 3/20 (15%). Regarding the biological specimens analyzed, the most frequent were bile (16 studies), gastric tissue (3 studies), liver tissue (5 studies), gallbladder tissue (4 studies), serum (7 studies), biliary epithelium (2 studies), gallstones (2 studies) and feces (1 study). Invasive procedures to obtain samples were endoscopic retrograde cholangiopancreatography (ERCP) (10 studies), percutaneous transhepatic cholangiography (PTC) (5 studies), surgery (liver resection, cholecystectomy) (10 studies) and liver biopsy (1 study).

#### 3.3.1. *Helicobacter* spp. under Investigation

The majority of the studies aimed to isolate any *Helicobacter* spp. (14 studies, *n* = 2064). However, some studies focused on specific species such as *Helicobacter pylori* (7 studies, *n* = 1453), *Helicobacter hepaticus* (2 studies, *n* = 322), *Helicobacter bilis* (2 studies, *n* = 116) or both *H. hepaticus* and *H. bilis* (1 study, *n* = 125).

#### 3.3.2. Controlling for Confounders and Risk of Bias Assessment

Fourteen studies comprising 1132 participants did not implement any methodology to control for demographic features, clinical characteristics, or laboratory parameters as potential confounders. Two studies were controlled by age and sex (*n* = 707), and two studies were controlled by age, sex, and body mass index (*n* = 293). Other studies attempted to control for a myriad of possible confounders, such as hepatitis B chronic infection, hepatitis C chronic infection, liver cirrhosis, smoking habit, alcohol consumption, coffee consumption, drug abuse, hemochromatosis, diabetes, cholesterol level, and educational level. The risk of bias assessment is presented in Table 2. None of the studies obtained a low risk of bias (A category) in all domains. In 16 out of 26 studies, at least one domain showed a high risk of bias (D category). The remaining 10 studies were considered at intermediate risk of bias with 2 or 3 domains classified in B or C categories. 

#### 3.3.3. Meta-Analysis

Seventeen studies (65.4%) reported a statistically significant association between *Helicobacter* infection and hepatobiliary tract abnormalities. Among the remaining nine studies, two studies did not report any patients with *Helicobacter* spp. infection, neither in patients with cancer nor among controls, [19,24] and seven studies did not perform a statistical comparison between cases and controls. In the meta-analysis including 18 studies and 1895 individuals (933 patients with cancer and 962 controls), the presence of *Helicobacter* spp. was associated with a significant increase in the risk of hepatobiliary tract malignancies (OR = 3.61 [95% CI 2.18–6]; *p* < 0.0001), with heterogeneity I^2^ = 61% across studies (*p* = 0.0003) (Figure 2). The funnel plot suggested a possible publication bias in this analysis (Figure 3). Regarding geographic distribution, 12 studies in the meta-analysis were performed in Asia (*n* = 671), obtaining OR = 3.67 (95% CI 1.92–6.99) (*p* < 0.001), and five studies took place in Europe (*n* = 1028), obtaining OR = 4.86 (95% CI 1.5–15.7) (*p* < 0.001). There was only one study from America in the meta-analysis. Heterogeneity (I^2^) was 38% in Asia (*p* = 0.09) and 80% in Europe (*p* = 0.008).

The effect of *Helicobacter* spp. on the risk of hepatobiliary tract malignancies was consistent irrespective of the type of specimen analyzed (Figure 4). In five studies, including 400 individuals with helicobacter assessed in bile specimens, the effect was OR = 3.57 (95% CI 1.73–7.39); *p* = 0.0006. Among two studies with 33 participants who underwent *Helicobacter* assessment in gastric tissue, the effect was OR = 42.63 (95% CI 5.25–346.24); *p* = 0.0004. In seven studies with 404 individuals with helicobacter assessed in the liver and/or biliary tissue specimens, the effect was OR = 4.92 (95% CI 1.90–12.76); *p* = 0.001. Finally, among three studies including 1009 participants with *Helicobacter* assessed in serum, the effect was OR = 1.38 (95% CI 1.00–1.90); *p* = 0.05. Heterogeneity was low (I^2^ ranging from 0% to 27%) for bile, gastric tissue, and serum (*p* = 0.24, *p* = 0.83, and *p* = 0.59, respectively). However, there was a significant heterogeneity (I^2^ = 57%; *p* = 0.02) when *Helicobacter* was assessed in liver/biliary tissue specimens.

## 4. Discussion

*Helicobacter* spp. may cause chronic inflammation in different organs and tissues, including the hepatobiliary tract [40]. In some types of malignancies, such as gastric cancer, the eradication of HP has been shown to prevent the disease in high-risk populations [41]. In this systematic review and meta-analysis, we showed a direct association between *Helicobacter* spp. and hepatobiliary tract malignancies, providing relevant data to enable future screening strategies and public health interventions in order to minimize this important health burden. 

HP infection is a major global health issue, but there is still little information available on the epidemiology, geographic distribution, and clinical impact of other *Helicobacter* species [42]. Regardless of its etiology, chronic inflammation and cholestasis are paramount factors in cholangiocarcinogenesis [43]. Chronic inflammation results in an increased exposure of cholangiocytes to interleukin-6, Tumor Necrosis Factor-ɑ, Cyclo-oxygenase-2, and other pro-inflammatory mediators, resulting in gradual mutations in tumor suppressor genes, proto-oncogenes and DNA mismatch-repair genes [44,45,46]. *Helicobacter* spp. in bile or liver tissue promotes chronic hepatitis in animal models and humans [25,47,48], enhances tissue inflammation in models of acute pancreatitis [49], induces mesenchymal transition in human cholangiocytes in vitro [50], and even hepatocellular and bile tract malignancies in genetically modified mice [51]. 

On the other hand, a complex interplay exists between *Helicobacter* spp. and liver fluke in endemic countries, where the latter infection could explain part of the carcinogenic effect [52,53]. *Opistorchis viverrini* is a well-established risk factor for CCA [54], and infected worms may be a reservoir for HP and HB in the biliary tree [55]. In the present review, two studies included data from *O. viverrini* co-infection, proving an increased risk for bile tract malignancies when both infections occur [34,36]. A synergistic pro-oncogenic effect of these pathogens is likely in endemic areas, but most of the studies did not report data about *O. viverrini* co-infection, and therefore we were unable to measure the extent of its interaction with *Helicobacter* species.

The presence of *Helicobacter* spp. in bile is able to promote lithiasis [56,57] and, in turn, intrahepatic and gallbladder lithiasis is able to increase the risk of bile duct malignancy and gallbladder cancer [57,58,59]. In this systematic review, ten studies included patients with cholelithiasis in the control group, which could have artificially reduced the expected effect of the infection by *Helicobacter* spp. alone. In fact, a meta-analysis published in 2014 tried to clarify this by avoiding studies whose controls presented cholelithiasis and concluded that there was a possible association between *Helicobacter* spp. infection and CCA [10]. Wang Y et al. [60] isolated HP proteins localized into the host cell nucleus of human GB cells and proposed that these could alter the normal function of GB cells, thus triggering carcinogenesis. Two of the analyzed studies included patients with gallbladder polyps, gallbladder adenomyomatosis, or chronic cholecystitis as controls, even when these abnormalities could be considered risk factors for GBC. 

Some studies have highlighted not only the possible relation between *Helicobacter* spp. and GBC [61,62] but also its relation with pancreatic cancer [63,64]. In addition, different meta-analyses have described *Helicobacter* as a possible risk factor for BTC [6,10]. Compared to previously published meta-analyses on the topic, the most recent including studies from 2001–2018 [6] and the previous one from 1990–2012 [10], our meta-analysis included studies from inception updated to 2022 and performed sub-analyses of different biological specimens, thus providing novel and interesting insights of diagnostic yield for screening purposes. In addition, we considered all Helicobacter species as some of them are enterohepatic and were not included in previous reports. 

One of the main sources of heterogeneity in this meta-analysis was the specimen used for the detection of *Helicobacter* spp., which varied among studies and was usually combined for improved accuracy. Serum may be the most appealing specimen due to its accessibility. However, the extrapolation of serum sample results to the presence of the bacteria in the biliary tract or hepatic tissue is a hard assumption. In our meta-analysis, the presence of *Helicobacter* spp. in serum obtained the weakest association with hepatobiliary tract malignancies, probably related to the above-referred limitation. On the other hand, bile and gastric tissue specimens were those showing the strongest association with hepatobiliary tract malignancies without a meaningful heterogeneity in the analysis. These specimens are accessible via endoscopy. Since patients with biliary lithiasis are at higher risk of hepatobiliary tract malignancies, bile samples could be obtained routinely if the patient undergoes endoscopic retrograde cholangiopancreatography for any reason, for instance, choledocholithiasis, indicating *Helicobacter* eradication therapy if the test results positive. There is insufficient information to rely on gastric tissue alone since we could only analyze two studies with 33 participants. Finally, direct detection in the tissue would definitely prove the presence of the bacteria in a non-expected organ, such as the pancreas or liver, that may be accessed by the migration of the bacteria through the bile tract [65]. 

Nowadays, several direct microbiological techniques for Helicobacter are available, among which polymerase chain reaction (PCR), immunohistochemical (IHC), enzyme-linked immunoassay (ELISA), specific staining, and culture are the most frequently used. The lowest diagnostic yield is obtained with culture techniques as these bacteria are very difficult to grow. Not dedicated stains such as hematoxylin and eosin (H&E), Giemsa, and Warthin-Starry silver may not be appropriate. The ELISA-based approach to identify *Helicobacter* spp. It could increase the identification rate, but it lacks specificity due to a potential cross-reactivity between *Campylobacter* and *Helicobacter* species. PCR offers higher sensitivity and specificity to detect *Helicobacter* spp. and is more feasible compared to the other methods, thus forming the gold standard for *Helicobacter* spp. diagnosis [66]. Certain genes of *Helicobacter* spp. could be targeted by PCR, including the 16S rRNA gene, the 26 K species-specific antigen gene, the glmM gene, ureA, ureB, cagA, and vacA genes. The most sensitive and widely used in this setting is 16S rRNA, which is a housekeeping gene genus-specific present in all bacterial species. Nonetheless, DNA damage could occur during tissue processing and therefore alter the results obtained by PCR. In addition, *Helicobacter* spp. present high mutation rates, which affect amplification efficiency, and false negative results could be expected [67]. There is also a geographic distribution of *Helicobacter* mutations, those affecting HP the best characterized in this setting [68]. 

Therapeutic eradication of *Helicobacter* spp. in subgroups of patients at increased risk of hepatobiliary tract malignancies with proven bile infection would be tremendously attractive in public health terms since there is no proven screening or intervention for these highly lethal diseases. Of note, none of the studies included in the analysis reported which measures were taken in those patients who tested positive for the bacteria, if any. Another important fact was that none of the studies reported if the patients had prior antibiotic exposure to eradicate HP, which is commonly isolated in the general population, especially in gastritis, duodenitis, and functional dyspepsia [69]. Future studies are required to determine if the same antibiotic approach commonly used for HP eradication would be effective in eliminating *Helicobacter* spp. from the bile and which tests could be relied on to ensure post-antibiotic eradication.

Our meta-analysis is not without limitations. The quality of the evidence was low, meaning a significant uncertainty of the results and vulnerability to bias. In addition, there is indirect evidence suggesting a possible publication bias. Included studies were heterogeneous in terms of the *Helicobacter* spp. and specimen analysed, geographical area, and controls used, who often had benign biliary diseases. These discrepancies have been translated into significant heterogeneity in the studies. However, the methodology used provided the strongest and most reliable evidence possible in the field, identifying the knowledge gaps for future study design. 

## 5. Conclusions

Based on low-quality evidence from observational retrospective case-control studies, *Helicobacter* spp. chronic infection detected by direct microbiological methods is associated with a meaningful increase in the risk of hepatobiliary tract malignancies. PCR analysis on bile specimens would be the optimal approach to screen for *Helicobacter* spp. considering accessibility, the strength of the association, and heterogeneity. Prospective studies are needed to ascertain which species are specifically associated with the pro-oncogenic risk and to determine if public health interventions on high-risk subpopulations would be cost-effective. 

## Figures and Tables

**Figure 1 cancers-15-00595-f001:**
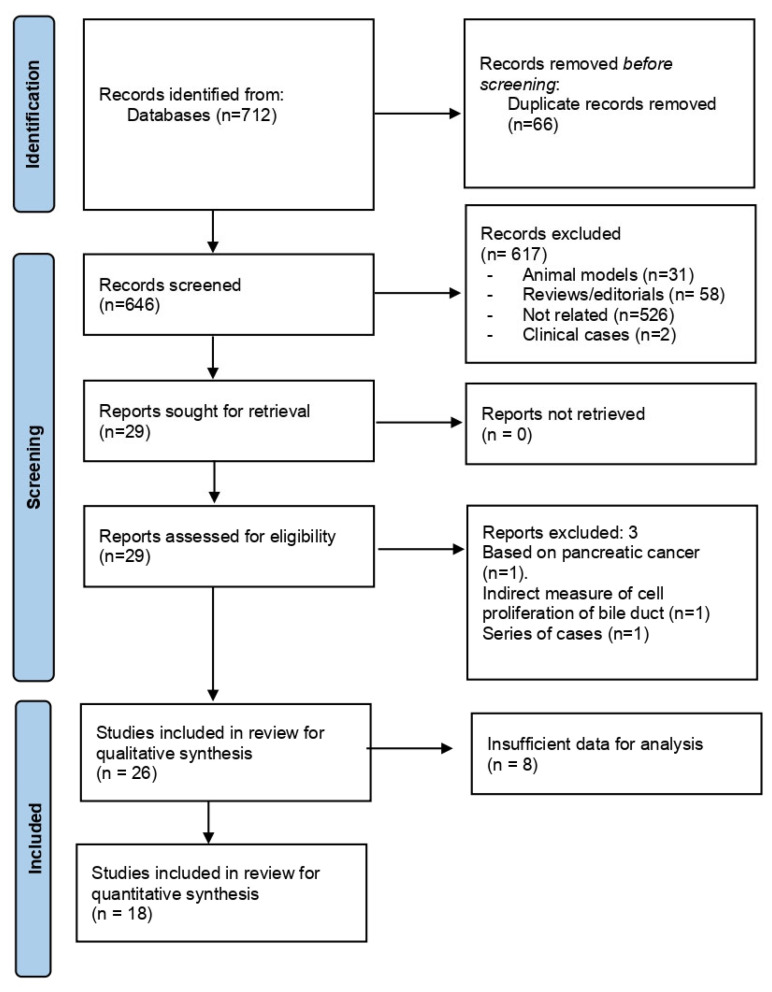
Flow diagram of the literature search and study selection.

**Figure 2 cancers-15-00595-f002:**
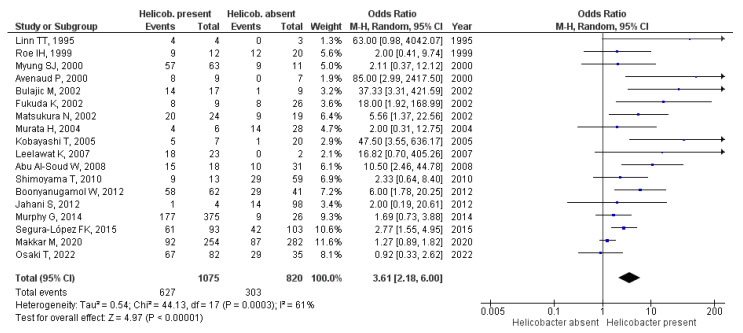
Forest plot of 18 studies evaluating the risk of hepatobiliary tract malignancies according to the presence or absence of *Helicobacter* spp. infection detected by direct microbiological methods using a random-effects model.

**Figure 3 cancers-15-00595-f003:**
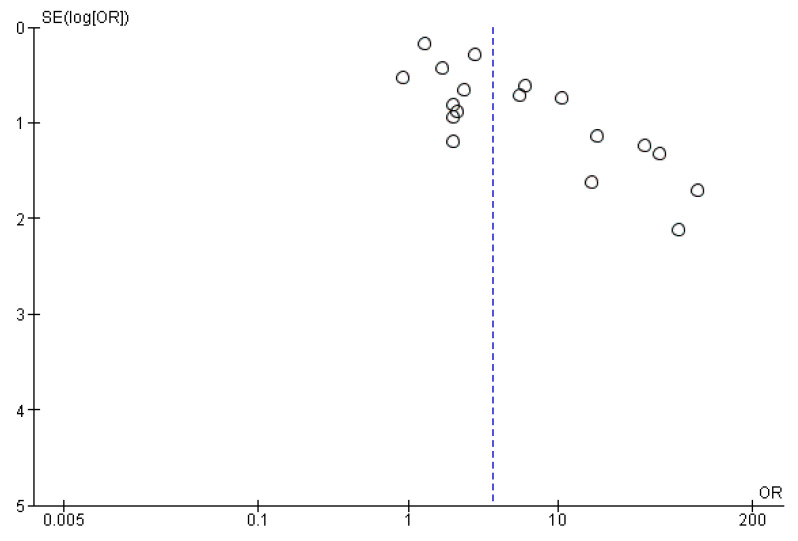
Funnel plot of 18 studies, including evaluating the risk of hepatobiliary tract malignancies according to the presence or absence of *Helicobacter* spp. infection detected by direct microbiological methods using a random-effects model.

**Figure 4 cancers-15-00595-f004:**
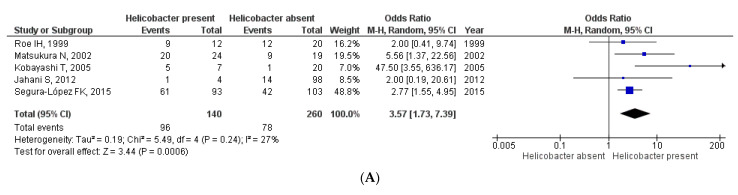
Forest plots of studies evaluating the risk of hepatobiliary tract malignancies according to the presence or absence of *Helicobacter* spp. infection assessed in different types of specimens: panel (**A**): bile; panel (**B**): gastric tissue; panel (**C**): liver/biliary tissue; panel (**D**): serum.

**Table 1 cancers-15-00595-t001:** Characteristics of the studies included in the systematic review.

Study	Year	Country	Specimen Type	Detection Method	Type of Malignancy in Case Group	Control Group	*H.* spp. Type	*H.* spp. + in Case Group	*H.* spp. + in Control Group	Results
Lin TT et al. [13]	1995	Taiwan	Gastric tissue & bile	PCR & CLOtest	PC, CCA, GC	Chronic cholecystitis	HP	PCR: 4/6 CLOtest 0/6	PCR 0/1 CLOtest 0/1	NR
Roe IH et al. [14]	1999	Korea	Bile	PCR & culture	CCA & PC	GS	*H.* spp.	PCR 9/21 Culture 0/21	PCR 3/11 CLOtest 0/11	NR
Myung SJ et al. [15]	2000	Korea	Bile, biliary tissue, GS & serum	PCR, ELISA, CLOtest & histology	G1: Hepatolithiasis, CCA, BS, papilomatosis, cystoadenocarcinoma. G2 (cholecystectomies) GS & GB polyp	GC	HP	Serum G1: 37/43 Serum G2: 20/23 PCR G1: 7/43 PCR G2: 0/23 CLOtest & histology G1 0/43, G2 0/23	Serum 6/8 Histopathology and CLOtest 0/8	NR
Avenaud P et al. [16]	2000	France	Liver tissue	PCR, culture & histology	HCC, CCA, hepatoCCA	PSC, Caroli, Hepatocellular adenoma, focal nodular hyperplasia	*H.* spp.	PCR 8/8 (7/8 H.Pylori & 1/8 H. Felis) Culture & histology 0/8	PCR 1/8 Culture & histology 0/8	NR
Bulajic M et al. [17]	2002	Yugoslavia	Gastric tissue & bile	PCR & CLOtest	BTC (GBC, CCA, Klatskin) & GS	PBC, Caroli, no stones/malignancy or not specified	*H.* spp.	PCR BTC 14/15 PCR GS 52/63 CLOtest BTC 12/15 CLOtest GS 37/63	PCR 3/11 CLOtest 4/11	BTC OR 9.9 (1.4–70.5)
Fukuda K et al. [18]	2002	Japan	Liver tissue & bile	PCR & IHQ	CCA, GBC & AoVC	GS, GB polyps, GB adenomyomatosis,	*H.* spp.	PCR bile 6/17 PCR tissue 8/16 IHQ 0/0	PCR 1/19 IHQ 3/19	BTC & *H.* spp. F value 5.05 (*p* = 0.041)
Matsukura N et al. [8]	2002	Japan & Thailand	Bile	PCR	Japan group: GS, BTC (CCA, GBC) Thai group: GS, BTC (CCA, GBC)	Non biliary disease	*H. bilis*	Japan: GS 9/16 Japan: BTC 13/15 Thai: GS 10/26 Thai: BTC 11/14	Japan 4/14	Japanese BTC OR 6.5 (1.09–38.63) Thai BTC OR 5.86 (1.21–26.33) Overall OR 6.4 (2.05–20.03)
Fallone CA et al. [19]	2003	Canada	Bile	PCR	CCA, AoVC, PC	GS, biliary stricture, pancreatitis, PSC & others not specified	*H.* spp.	0/15	0/110	NR
Murata H et al. [20]	2004	Japan	Gallbladder tissue	PCR	GBC, CCA, PC	GS	*H. bilis*	4/18	2/16	NR
Kobayashi T et al. [21]	2005	Japan	Bile	PCR & culture	BTC (GBC, CCA), GS	Non biliary disease	*H.* spp.	BTC: 5/6 GS 16/30	2/21	BTC vs. GCC *H.* spp. *p* < 0.05
Tiwari SK et al. [22]	2006	India	Gastric tissue & bile	PCR, CLOtest & culture	Hepatobiliary disorder (CCA, PC & others)	Gastric disorders	HP	Bile 29/30 Gastric 26/30	Bile 2/30 Gastric 28/30	NR
Leelawat K et al. [23]	2007	Thailand	Liver tissue	PCR	HCC & CCA	Liver metastasis from CRC, IH duct stones	HP VacA	18/18	5/7	NR
Bohr URM et al. [24]	2007	Germany	Gallbladder tissue	PCR, culture & IHQ	GBC & GB disease	Cholecystectomy	*H.* spp.	GBC 0/20 GBD 1/57 IHQ 0/77 Culture 0/77	0/22	NR
Al-Soud WA et al. [25]	2008	Sweden	Liver tissue	PCR	HCC & CCA	Hepatocellular adenoma, focal nodular hyperplasia, fatty lesion & hematoma	*H.* spp.	HCC 7/12 CCA 8/13	3/24	*p* < 0.01
Shimoyama T et al. [26]	2010	Japan	Serum	Western Blot	GS, BTC (CCA, GBC), PC	Patients undergoing endoscopy with no gastric ulcer or cancer	*H. hepaticus*	GS 11/55 BTC 7/18 PC 2/19	4/34	BTC vs. controls *p* < 0.05
Murakami K et al. [27]	2011	Japan	Serum	ELISA	Liver disease, Upper GI disease, Lower GI disease, biliary tract disease, pancreas disease	Healthy blood donors	*H. hepaticus*	LD 34/69 UGID 6/38 LGID 1/17 BTD 6/26 PD 3/16	8/30	H.hepaticus in LD vs. all the other groups *p* < 0.05 for each comparison
Yakoob J et al. [28]	2011	Pakistan	Gallbladder tissue & bile	PCR	GB polyps & GBC	Chronic cholecystitis	*H.* spp.	5/55	28/89	*p* = 0.03
Boonyanugamol W et al. [29]	2012	Thailand	Bile, liver tissue, gallbladder tissue & gallstones	PCR & culture	CCA, GS, CCA, PC	Autopsies	*H.* spp.	CCA 58/87 GS 22/53	4/16	*p* < 0.05
Jahani S et al. [30]	2012	Iran	Bile	PCR & culture	BTC, PC	Gallstones, other diseases not cancer	*H.* spp.	PCR 1/15 Culture 0/15	PCR 3/87	NR
Murphy G et al. [31]	2014	Finland	Serum	Multiplex assay	BTC (GBC, CCA, AoVC), LC (IHDC, HCC)	Patients from ATBC study *	*H.* spp.	HP + BTC: 62/64 LC: 115/122	198/224	OR for overall cases 2.63 (1.08–6.37) OR for BTC 5.47 (1.17–25.65)
Segura-López FK et al. [32]	2015	Mexico	Bile	PCR	CCA, AoVC, GBC	Benign biliary pathology	*H.* spp.	H.Bilis 44/103 H.Hepaticus 17/103	H.Bilis 19/91 H.Hepaticus 13/91	*H. bilis* & CCA OR 2.83(1.49–5.32) *H. hepaticus* & CCA OR 1.19 (0.54–2.60)
Aviles-Jiménez et al. [33]	2016	Mexico	Bile	PCR	BTC	Benign biliary pathology	HP	VacA &/or CagA+ 75/100 VacA+ 50/97 CagA+ 46/100	VacA&/or CagA + 52/92 VacA+: 21/86 CagA+: 39/92	HP + *p* = 0.035 VacA + *p* = 0.0003
Deenonpoe R et al. [34]	2017	Thailand	Feces	PCR	OV + patients	OV- patients	*H.* spp.	HP+: 190/293 H.Bilis: 86/293 Both HP + HB: 79/293	HP+: 77/260 H.Bilis 14/260 Both HP + HB: 10/260	*H.* spp. more freq in OV+ *p* < 0.0001 Advanced periductal fibrosis in OV + HP+ cagA RRR 3.38 (1.52–7.58)
Makkar M et al. [35]	2020	Germany	Serum	PCR	CCA, HCC	Patients from the PLCO trial with no cancer **	HP	CCA 35/74 HCC 57/105	162/357	HP CagA+ HCC OR 1.96 (1.21–2.18) HP CagA + BTC OR 2.16 (1.03–4.50)
Jala I et al. [36]	2021	Thailand	Serum	ELISA	OV + with no HB abnormalities; OV+ with HB abnormalities, OV+ CCA	OV- patients with no HB abnormalities	HP	pCagA+: OV+ no HBA: 112/140 OV+ HBA: 119/144 OV + CCA: 117/145 GroEL+ OV+ no HBA: 98/140 OV + HBA: 117/144 OV + CCA: 118/145	pCagA+: 98/122 GroEL+ 82/122	HP + OV + OR for HBA 2.11 (1.20–3.71) OR for CCA 2.13 (1.21–3.75)
Osaki T et al. [37]	2022	Japan	Serum, bile, and biliary tissue	PCR, ELISA & culture	BTC (CCA, AoVC, GBC), PC	Cholelythiasis, GC, CRC & other suspected cancers not specified	*H. hepaticus* & *H. bilis*	PCR *H. bilis* BTC 2/35; PC 6/59 PCR *H. hepaticus* BTC 2/35; PC 1/59 ELISA *H. bilis* BTC 13/37; PC 24/59 ELISA *H. hepaticus* BTC 11/37; PC 19/59	PCR *H. bilis*: 3/21 PCR *H. hepaticus* 1/21 ELISA *H. bilis* 8/21 ELISA *H. hepaticus* 7/21	HB+ and HH+ PC vs. controls *p* = 0.046

AoVC: Ampulla of Vater cancer; BS: Benign stricture; BTC: biliary tract cancer; CCA: cholangiocarcinoma; CRC: colorectal cancer; GB: gallbladder; GBC: gallbladder cancer; GC: gastric cancer; GS: Gallstones; G1: group 1. G2: group 2; HepatoCCA: Hepato-cholangiocarcinoma.; HCC: hepatocellular carcinoma; HP: *Helicobacter pylori*; IH: intrahepatic; IHDC: intrahepatic duct cancer; IHQ: immunohistochemical; LC: liver cancer; NR: not reported; OV+: *O. viverrini* positive; PBC: primary biliary cholangitis; PC: pancreatic cancer; PSC: primary sclerosing cholangitis. * ATBC study: Alpha-Tocopherol Beta-Carotene Cancer Prevention Study is a randomized, double-blind, placebo-controlled trial designed to determine whether daily supplementation with a-tocopherol, b-carotene, or both, would reduce the incidence of lung cancer in male smokers [38]. ** PLCO trial: Prostate, Lung, Colorectal, and Ovarian Cancer Screening Trial was designed be conducted to assess whether screening exams reduce mortality for prostate, lung, colorectal, and ovarian cancers [39].

**Table 2 cancers-15-00595-t002:** The checklist provided by the CLARITY group at McMaster University evaluates the risk of bias to assess the quality of case-control studies.

Study	Assessment of Exposure	Outcome of Interest in Cases but Not Controls	Cases Selection	Controls Selection	Case-Control Matching and Adjusted for Confounders
Lin TT et al. [13]	A	C	D	C	D
Roe IH et al. [14]	A	C	C	B	D
Myung SJ et al. [15]	A	C	B	A	D
Avenaud P et al. [16]	A	B	B	C	D
Bulajic M et al. [17]	A	B	A	B	B
Fukuda K et al. [18]	A	B	A	C	C
Matsukura N et al. [8]	A	B	A	C	D
Fallone CA et al. [19]	A	B	A	C	D
Murata H et al. [20]	A	B	A	C	D
Kobayashi T et al. [21]	A	B	A	B	D
Tiwari SK et al. [22]	A	C	A	C	D
Leelawat K et al. [23]	B	B	A	C	D
Bohr URM et al. [24]	A	B	A	B	B
Al-Soud WA et al. [25]	A	A	A	B	D
Shimoyama T et al. [26]	B	B	A	B	D
Murakami K et al. [27]	B	C	C	B	D
Yakoob J et al. [28]	A	C	B	C	D
Boonyanugamol W et al. [29]	A	B	A	B	D
Jahani S et al. [30]	A	B	A	C	D
Murphy G et al. [31]	B	A	A	A	B
Segura-López FK et al. [32]	A	B	A	B	B
Aviles-Jiménez et al. [33]	A	B	A	B	B
Deenonpoe R et al. [34]	B	A	B	B	B
Makkar M et al. [35]	B	A	A	B	A
Jala I et al. [36]	B	A	A	A	B
Osaki T et al. [37]	A	B	A	C	C

A: Definitely yes (low risk of bias), B: Probably yes, C: Probably no, D: Definitely no (high risk of bias).

## Data Availability

Data will be available upon reasonable request.

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
