# Peer review of "Helicobacter Species and Hepato-Biliary Tract Malignancies: A Systematic Review and Meta-Analysis"

_cancers, 2023, doi:10.3390/cancers15030595_

Round 1

Reviewer 1 Report

I thank the authors for writing this interesting manuscript. Unfortunately but this is related to the field, this topic is difficult to summarize as studies came from different countries, are focused on different Helicobacter ss, different cancers and used inadequate number of patients / controls. 

Perhaps splitting analysis following geographic area could be useful as it is obvious that, at least for biliary tract cancers, epidemiological data are different!

Under some more precise comments

Review of the manuscript :

« Helicobacter Species and biliary tract malignancies : a systematic review and meta-analysis »

Gros et al. Cancers 2088824

                This article reports a systematic review and meta-analysis to evaluate the role of Helicobacter species infection and « biliary tract » cancers.

Major criticisms

The title is not in accordance with the text : in fact this review is more focused on Hepato-biliary tract malignancies.

Table 2 : difficult to understand, in all cases but one, the * were between B and C score …

Minor comments

Abstract : instead of (n=4,034) write (4 ,034 cases or patients)

Line 73 : helicobacter

Description of literature search results and Figure 1 are a little bit different.

Line 186 : the total is 26 studies, while only 25 (qualitative synthesis) or 17 (quantitative) were included ; same inconsistency line 207-209 : 16 studies … among the remaining 11 studies …

In the discussion part insist more on the large discrepencies between these different studies illustrated by heterogeneity!

Author Response

#TO THE REVIEWER 1:

* Original comment: “Perhaps splitting analysis following geographic area could be useful as it is obvious that, at least for biliary tract cancers, epidemiological data are different!”

Answer: We have added a new analysis of studies performed in Asia and in Europe sepparately. There was only one study form America for quantitative synthesis, thus precluding meta-analysis. The included paragraph reads as follows: “Regarding geographic distribution, 12 studies in the meta-analysis were performed in Asia (n=671), obtaining OR=3.67 (95%CI 1.92-6.99) (p<0.001), and 5 studies took place in Europe (n=1,028), obtaining OR=4.86 (95%CI 1.5-15.7) (p<0.001). There was only one study from America in the meta-analysis. Heterogeneity (I2) was 38% in Asia (p=0.09) and 80% in Europe (p=0.008).

* Original comment: “The title is not in accordance with the text: in fact this review is more focused on Hepato-biliary tract malignancies”.

We completely agree. We have modified the title as per reviewer’s suggestion.  

* Original comment: “Table 2 is difficult to understand, in all cases but one, the * were between B and C score…”

Answer: We apologize because it seems that the table had a formatting issue, making its interpretation difficult. In the new version, this table was modified to avoid misunderstanding.

* Original comment: “Abstract: instead of (n=4,034) write (4,034 cases or patients)”

Answer: Thank you for the appreciation. We have implemented this modification in the abstract.

* Original comment: “Line 73: helicobacter”

Answer: This typo has been corrected.

* Original comment: “Description of literature search results and Figure 1 are a little bit different”.

Answer: Thank you for the appreciation. We have carefully revised the data in figure 1 and had made appropriate corrections.

* Original comment: “Line 186: the total is 26 studies, while only 25 (qualitative synthesis) or 17 (quantitative) were included; same inconsistency line 207-209 : 16 studies … among the remaining 11 studies …”

Answer: We have carefully checked these numbers and made appropriate corrections taking into account that an additional study, which was rightly identified by one of the reviewers, has been included in the new version of the manuscript. There are 26 studies included for qualitative synthesis and 18 studies qualified for meta-analysis.

* Original comment: “In the discussion part insist more on the large discrepancies between these different studies illustrated by heterogeneity!”

Thanks for your comment. We have added a comment in the limitation’s paragraph which reads as follows: “Included studies were heterogeneous in terms of the Helicobacter spp. and specimen analysed, geographical area, and controls used, who often had benign biliary diseases. These discrepancies have been translated into significant heterogeneity of the studies”.

Reviewer 2 Report

In their manuscript, Gros et al. performed a meta-analysis of studies from inception to date to identify a possible correlation between Helicobacter species and biliary tract malignancies. The authors concluded that chronic infection with Helicobacter species is associated with a significant risk of hepatobiliary tract malignancy. The concept of this review is novel and relevant, as the authors included studies from inception to date and highlighted the need for attention in this area. They also suggest PCR as the only relevant method for detection. The manuscript is well-written and relevant studies have been cited, but a few concerns that need to be addressed.

The authors have discussed the mutation of helicobacter in BTC as an issue for its detection; however, to further strengthen their findings, they may consider discussing the potential impact of geographical factors on the mutation of helicobacter in BTC. Furthermore, including a discussion of additional potential limitations of the PCR detection method, such as the impact of DNA damage due to tissue processing, would provide a more comprehensive and robust analysis of the results.

The Results section is detailed, however, providing references for the geographical locations of the studies in the 'Search Results and Study Characteristics' would provide more context. Additionally, the authors should explain the omission of studies from Sweden (PMID: 18083084) and Yugoslavia (12404289).

In lines 171-172, it is unclear if the authors intend to include additional methods in the list of ‘surgery (liver resection, cholecystectomy…) (9 studies)’ in the results section.

The authors have included statistical analysis within the Data Extraction and Risk of Bias Assessment subsection of the method section, however, adding the term "statistical analysis" in the title of the subsection will aid in clearly identifying the methods used.

Author Response

#TO THE REVIEWER 2:

* Original comment: “The authors have discussed the mutation of helicobacter in BTC as an issue for its detection; however, to further strengthen their findings, they may consider discussing the potential impact of geographical factors on the mutation of helicobacter in BTC. Furthermore, including a discussion of additional potential limitations of the PCR detection method, such as the impact of DNA damage due to tissue processing, would provide a more comprehensive and robust analysis of the results.”

We agree with the reviewer. We have extended our text in the discussion to cover this issue, which reads as follows: “Nonetheless, DNA damage could occur during tissue processing and therefore alter the results obtained by PCR. In addition, Helicobacter spp. present high mutation rates which affect amplification efficiency and false negative results could be expected [67]. There is also a geographic distribution of Helicobacter mutations, being those affecting HP the best characterized in this setting[68]”. We added a new reference which analysed Helicobacter pylori genome distributions according to geographic areas (doi:10.3389/fmicb.2021.687259).

* Original comment: “The Results section is detailed, however, providing references for the geographical locations of the studies in the 'Search Results and Study Characteristics' would provide more context. Additionally, the authors should explain the omission of studies from Sweden (PMID: 18083084) and Yugoslavia (12404289).”

Answer: Thank you for your careful evaluation. One of the studies suggested by the reviewer (PMID: 12404289) was included in the previous version of the manuscript but it was wrongly attributed to Germany instead of Yugoslavia (As Serbia became and independent country in 2006 and the study was published in 2002 we have kept Yugoslavia). We have corrected this in the table. The second study from Sweden (PMID: 18083084) was overlooked in our literature search and it has been now included both in qualitative and quantitative synthesis.

* Original comment: “In lines 171-172, it is unclear if the authors intend to include additional methods in the list of ‘surgery (liver resection, cholecystectomy…) (9 studies)’ in the results section.”

Answer: To avoid misunderstanding we have removed the “…” as there were no other types of surgeries.

 * Original comment: “The authors have included statistical analysis within the Data Extraction and Risk of Bias Assessment subsection of the method section, however, adding the term "statistical analysis" in the title of the subsection will aid in clearly identifying the methods used.”

Answer: We have included a new subheading in methods as suggested as follows: “2.4. Statistical analysis and Risk of Bias Assessment”.

Reviewer 3 Report

Summary

The review and meta-analysis is dealing with the connection between chronic Helicobacter subspecies infection and the connection to development of biliary tract cancer. The manuscript has been submitted to the topical collection “Targeting solid tumors”. With 25 studies included in qualitative synthesis and 17 were eligible for meta-analysis. The data was heterogeneous, and the overall quality of evidence was low. Nevertheless, a significant increase of risk for biliary tract cancer could be shown for patients with proof of Helicobacter species.

Comments

·      Table 2 seems to have a formatting problem with all marks between B/C. I guess the table should show the risk of bias for every parameter and study. This must undergo a workover.

Conclusion

The manuscript is worked out and written well. The review and meta-analysis are performed with conscientiousness and according to the scientific fundamental rules. The authors communicate the heterogenous data and the high to intermediate risk of bias honestly and are able to show the potential and within this meta-analysis significant influence of chronic Helicobacter spp. infection for development of biliary tract cancer.

There is not much to criticize. The manuscript suits good within the topical collection (Targeting solid tumors) and overall, the findings and presentation are good. From my point of view, it offers an interesting insight in a topic not much recognized, yet. I recommend minor changes (table 2) and accepting the manuscript afterwards for publication in the topical collection of Cancers.

Author Response

# TO THE REVIEWER 3

* Original comment: “Table 2 seems to have a formatting problem with all marks between B/C. I guess the table should show the risk of bias for every parameter and study. This must undergo a workover.”

Answer: We agree with the reviewer that this table had some formatting issues, which made it hard to understand some categories. We have modified the table to ease its interpretation.